# A Comprehensive Review on Neuroendocrine Neoplasms: Presentation, Pathophysiology and Management

**DOI:** 10.3390/jcm12155138

**Published:** 2023-08-05

**Authors:** Qamar Sultana, Jill Kar, Amogh Verma, Shreya Sanghvi, Nirja Kaka, Neil Patel, Yashendra Sethi, Hitesh Chopra, Mohammad Amjad Kamal, Nigel H. Greig

**Affiliations:** 1Department of Medicine, Deccan College of Medical Sciences, Hyderabad 500058, India; dr.sultanaq@gmail.com; 2PearResearch, Dehradun 248001, India; jillkmed@gmail.com (J.K.); amoghverma2000@gmail.com (A.V.); ssanghvi01@gmail.com (S.S.); nirja@pearresearch.com (N.K.); neil@pearresearch.com (N.P.); 3Department of Medicine, Lady Hardinge Medical College, New Delhi 110001, India; 4Rama Medical College Hospital and Research Centre, Hapur 245304, India; 5Lokmanya Tilak Municipal Medical College and General Hospital, Mumbai 400022, India; 6Department of Medicine, GMERS Medical College, Himmatnagar 390021, India; 7Government Doon Medical College, HNB Uttarakhand Medical Education University, Dehradun 248001, India; 8Chitkara College of Pharmacy, Chitkara University, Rajpura 140401, India; chopraontheride@gmail.com; 9Institutes for Systems Genetics, Frontiers Science Center for Disease-Related Molecular Network, West China Hospital, Sichuan University, Chengdu 610017, China; rrs.usa.au@gmail.com; 10King Fahd Medical Research Center, King Abdulaziz University, Jeddah 21589, Saudi Arabia; 11Department of Pharmacy, Faculty of Allied Health Sciences, Daffodil International University, Dhaka 1216, Bangladesh; 12Enzymoics, Hebersham, NSW 2770, Australia; 13Novel Global Community Educational Foundation, Hebersham, NSW 2770, Australia; 14Drug Design & Development Section, Translational Gerontology Branch, Intramural Research Program, National Institute on Aging, National Institutes of Health, Baltimore, MD 21224, USA

**Keywords:** neuroendocrine tumors, neuroendocrine carcinomas, carcinoid syndrome, pulmonary neuroendocrine tumors, gastroenteropancreatic neuroendocrine tumors

## Abstract

Neuroendocrine neoplasms (NENs) are a group of heterogeneous tumors with neuroendocrine differentiation that can arise from any organ. They account for 2% of all malignancies in the United States. A significant proportion of NEN patients experience endocrine imbalances consequent to increased amine or peptide hormone secretion, impacting their quality of life and prognosis. Over the last decade, pathologic categorization, diagnostic techniques and therapeutic choices for NENs—both well-differentiated neuroendocrine tumors (NETs) and poorly differentiated neuroendocrine carcinomas (NECs)—have appreciably evolved. Diagnosis of NEN mostly follows a suspicion from clinical features or incidental imaging findings. Hormonal or non-hormonal biomarkers (like serum serotonin, urine 5-HIAA, gastrin and VIP) and histology of a suspected NEN is, therefore, critical for both confirmation of the diagnosis and classification as an NET or NEC. Therapy for NENs has progressed recently based on a better molecular understanding, including the involvement of mTOR, VEGF and peptide receptor radionuclide therapy (PRRT), which add to the growing evidence supporting the possibility of treatment beyond complete resection. As the incidence of NENs is on the rise in the United States and several other countries, physicians are more likely to see these cases, and their better understanding may support earlier diagnosis and tailoring treatment to the patient. We have compiled clinically significant evidence for NENs, including relevant changes to clinical practice that have greatly updated our diagnostic and therapeutic approach for NEN patients.

## 1. Introduction

Neuroendocrine neoplasms (NENs) are rare abnormal growths that originate from widely distributed cells within the neuroendocrine system. They secrete peptide hormones and present with a broad spectrum of symptoms based on the hormone secreted. They vary largely in the extent of their metastatic pattern [1,2]. The resulting syndromes and associated hormonal dysregulation impact the well-being and prognosis of afflicted patients. Modern epidemiological trends (SEER program) suggest an increase in incidence, especially among females [3]. Their prevalence is estimated to be less than 200,000 in the United States [4]. In this regard, Yao and colleagues identified variations in race, gender and age upon diagnosis of these tumors [5]. At the current time, we know little about metastases from NENs at the population level [6].

Based on histology, NENs can be divided into two major types: well-differentiated neuroendocrine tumors (NETs) and poorly differentiated neuroendocrine carcinomas (NECs). Histologically, NETs comprise cells with oval to round nuclei and granular-looking chromatin that provides a “salt and pepper” appearance with a higher degree of expression of neuroendocrine markers. In contrast, NECs exhibit poor differentiation and present with a “sheetlike” growth with a lower degree of neuroendocrine marker expression [4]. The tumors exhibit a different morphology according to their differentiation state. Well-differentiated neuroendocrine tumors (NETs) are often well confined, sometimes encapsulated (in the pancreas) and tumors that have a uniform cut surface can occur anywhere in the digestive tract. Histologically, they are distinguished by the presence of an organoid proliferation of homogeneous cells, abundant granular and eosinophilic cytoplasm and a high number of secretory granules inside the cells. They demonstrate deep penetration of the intestine wall or the peripancreatic tissue (in the pancreas), and they have mostly metastasized when they are detected. Macroscopically, they are poorly defined, and they may show vast areas of necrosis and bleeding. In terms of their appearance under a microscope, PD-NECs are distinguished by a solid proliferation of cells that can be found either in big nests or in sheets with enormous “geographic chart” necrotic zones. However, NETs and NECs can show an overlap in their gross morphologies [7,8].

NETs and NECs can occur within almost any organ, and their most common site is the gastrointestinal tract (>60%)—most frequently in the “midgut”, followed by the “foregut” and then by the “hindgut” (Figure 1) [9,10]. The lung is the second most primary common site (>20%) [10]. Other sites include the head and neck, thymus, thyroid, breast, skin and genitourinary system [11,12,13,14], with some cases presenting from metastasis of an occult primary [14]. The presence of these tumors in extra-thoracic and extra-digestive organs is relatively rare [14]. Such tumors were referred to as “carcinoid tumors” during the past. They have either an indolent course (NETs) or an aggressive one (NECs) and are often accompanied by the clinical features of flushing, diarrhea and heart disease and with site-specific differential characteristics that include bronchospasm, myopathy, skin pigmentation and other paraneoplastic manifestations and syndromes [15]. The clinical features are majorly based on the type of NEN and the involved cell that secretes the hormone. The secretion of this hormone is regulated by complex mechanisms, as described in Figure 2.

Medical treatment options have become broader in recent years and are of variable efficacy [16]. These include cytokine therapy (e.g., interferon-α), vascular endothelial growth factor (VEGF) inhibitors, somatostatin analogs, mechanistic target of rapamycin (mTOR) inhibitors and radiotherapy [17]. Among these, several biologics that effectively target pathways upregulated in cancer cells have recently been approved [18,19,20]. Within this review, we have distilled the existing and evolving literature on NETs and NECs (PubMed and Scopus 1985 to May 2023) to comprehensively cover the classification, staging, genetics, common sites, clinical presentation and evolving diagnostic and therapeutic modalities. With this preface, our review focuses on providing clinical and basic research scientists with a comprehensive synopsis to aid their understanding of NENs. Overall, this condition has diagnostic issues, especially with overlapping and broad symptoms which do not necessarily point toward the diagnosis of NEN. Hence, the authors have attempted to compile and delineate the features that should prompt the clinician to suspect NEN. In general, a high clinical suspicion is mandated for the diagnosis of NEN.

**Figure 2 jcm-12-05138-f002:**
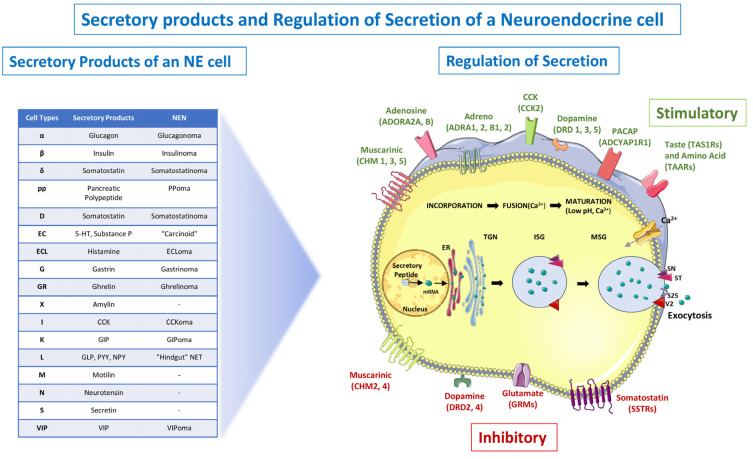
Secretory products and Regulation of Secretion of a Neuroendocrine cell. The nucleus and endoplasmic reticulum (ER) are the sites of initial transcription and processing, and secretory products aggregate in the trans-Golgi network (TGN). They are then integrated into immature vesicles together with other protein products destined for immature secretory vesicles (ISGs). Multiple ISGs fuse to form a mature secretory granule (MSG) by a process that includes calcium (Ca^2+^) influx, granule acidification, prohormone processing and amine uptake. Positive regulatory inputs from several regulatory G-protein coupled receptors (GPCRs) (green) direct this series of processes. Ligand binding causes both signaling pathways (PKA/cAMP, MAPK, PI3K/DAG/PKC) and membrane depolarization to be activated. Regulatory GPCRs, which include muscarinic, tastant and trace amine receptors, are often cell-type-specific. MSGs are directed to the plasma membrane as a result of activation, and docking occurs at the cell membrane following receptor-mediated Ca^2+^ influx. Syntaxin (SY), synaptotagmin (ST), vesicle-associated membrane protein 2 (VAMP2) (V2) and synaptosomal-associated protein, 25-kDa (SNAP25) (S25) are all expressed throughout this process (red and purple arrowheads). The resulting vesicle-and-membrane fusion process results in MSG release of contents into the extracellular space (exocytosis). Secretion is inhibited by a number of GPCRs (red) (somatostatin > muscarinic > glutamate), which, when activated, reverse the signaling pathway. Green dots in the figure represent secretory proteins. For each of the GPCRs, IUPHAR gene symbols are included. CCK—cholecystokinin; GIP—gastric inhibitory peptide; GLP-1—glucagon-like peptide 1; NPY—neuropeptide Y (tyrosine); PP—pancreatic polypeptide; PYY—polypeptide YY (tyrosine–tyrosine). *Figure credit: The content of the figure is derived from the open-access information and illustrations published by Kidd* et al. [21]. *Parts of the figure were generated by making use of pictures available from Servier Medical Art (smart.servier.com), accessed from Servier, and licensed under a Creative Commons Attribution 3.0 unported license*.

## 2. Classification and Staging

The classification of NEN has been updated multiple times over recent years with the latest update being the WHO Classification of Endocrine and NEN in 2022 [8,22]. Essentially, tumor cells that retain the molecular and morphological features of neuroendocrine cells, are well differentiated and are termed neuroendocrine neoplasms (NETs). These are divided into three grades: G1, G2 and G3. Tumor cells that have severe atypia of cells and abnormal molecular or genetic features and preservation of neuroendocrine markers are epithelial poorly differentiated and are called NECs [8]. An aggressive neoplasm that combines both a recognizable neuroendocrine and non-neuroendocrine part is labeled a mixed neuroendocrine–non-neuroendocrine neoplasm (MiNENs). The term MiNEN is in use for the first time within the new classification to define mixed neoplasms having their origin in all organ systems when the diagnostic criteria are met. NENs of non-endocrine organs were incorporated for the first time in the 2022 WHO Classification of Endocrine Tumors [8]. The diagnostic criteria for the classification of NENs are mainly based on two components of the cell cycle: the mitotic count, determined by the number of cells that are dividing observed in a specific area under a microscope (2 mm^2^), and the Ki-67 index, where Ki-67 is a protein that increases in amount as the cells prepare themselves to divide. In the event that cells in an area with Ki-67 fall under a high percentage, this indicates rapid division [23]. Classification for different anatomic sites is shown in Table 1.

Staging: The heterogeneous behavior of NENs makes it challenging to devise a practical staging system that can help by providing accurate prognostic information [24]. The European Neuroendocrine Tumor Society (ENETS) previously provided recommendations for tumor–node–metastasis (TNM) staging of gastroenteropancreatic NETs. The American Joint Committee on Cancer (AJCC) recently put forward a TNM staging manual that includes NETs of different anatomic sites [24,25,26]. For lung NETs, non-small cell lung cancer (NSCLC) staging criteria are applied, but there are recommendations that NETs of the lungs should be staged independently from NSCLCs; a tumor-specific system of staging for lung NETs must include the histologic grade [4,27].

## 3. Carcinoid Tumor NETs

Carcinoid tumors are a form of neuroendocrine tumors that can arise from the enterochromaffin cells located chiefly within the gastrointestinal and bronchopulmonary systems [1]. The term carcinoid originates from the carcinoma-like nature of these tumors [28]. Although initially thought to be benign, they display malignant properties like metastasis, lymphovascular invasion and recurrence after treatment. These cells are chromium-staining that release serotonin. A majority of carcinoid tumors do not present with carcinoid syndrome as the vasoactive hormones secreted by them are neutralized by the liver. Only the carcinoid tumors that metastasize to the liver escape this process and cause these vasoactive hormones to be released into systemic circulation, leading to a symptomatic presentation [29]. The secretion of serotonin is responsible for the classic carcinoid syndrome associated with flushing, bronchoconstriction, diarrhea and right-sided valvular disease [28]. The incidence of carcinoid tumors is increasing, which may partly be attributed to an improvement in diagnostic tools. Traditionally, carcinoid tumors were classified as per their embryological origin into (i) foregut (bronchopulmonary, thymus, stomach and duodenum), (ii) midgut (ileum, jejunum and proximal colon) and (iii) hindgut (distal colon and rectum) [4].

## 4. Pulmonary NENs

As per the WHO, pulmonary NENs are classified into typical carcinoids, atypical carcinoids, large cell NEC and small cell lung cancer based on their mitotic activity, presence of necrosis and cytology [30,31]. NENs account for 25% of all primary lung malignancies [32,33]. Diagnosis is dependent on the presence of certain immunohistochemical markers, such as chromogranin A, synaptophysin and neuron-specific enolase [33]. Carcinoids are typically found in younger patients (45–50 years old) without a gender preference or a history of tobacco use [4,34,35]. On the other hand, high-grade NETs are frequently seen in older males with a history of smoking [34,36,37]. Typical carcinoids are defined as tumors having <2 mitoses/2 mm^2^ and no necrosis. In contrast, atypical carcinoids have 2–10 mitoses/2 mm^2^ and/or the presence of necrosis [34,35]. Microscopically, SCLC appears as small blue cells with scanty cytoplasm, faint nucleoli, nuclear molding and more than 10 mitoses/2 mm^2^ [31,37]. Pulmonary large cell NEC (LCNEC) has similar features, except the cells have a low nucleus-to-cytoplasm ratio with prominent nucleoli [31].

The clinical features vary as per the location of the tumors. Typical carcinoids and SCLC are often centrally located and present with a cough, hemoptysis and recurrent pneumonia due to bronchial obstruction and tumor ulceration [30,34,35]. Atypical carcinoids and LCNEC are peripherally located and are usually seen as incidental findings on imaging. Patients may present with non-specific symptoms, like dyspnea, cough, anorexia and weight loss [4,30,35,36]. Compression and invasion of the surrounding tissues can lead to dysphagia, SVC syndrome or hoarseness of voice [34]. More than 90% of pulmonary NENs are non-functioning but SCLC can present with carcinoid syndrome, Cushing’s syndrome and syndrome of inappropriate antidiuretic hormone (SIADH) secretion [4,33,36]. Common sites of metastasis include the brain, liver, bones and adrenals [4].

A chest computerized tomography (CT) scan is the investigation of choice. Flexible bronchoscopy or CT-guided puncture biopsy is often needed for the histopathological confirmation of diagnosis. Modalities like an octreotide scan can be used to identify distant metastasis [33]. Typical carcinoids are generally treated with conservation procedures, like wedge resection, segmentectomy or sleeve resection. Bronchoscopic resection can be attempted in small, polypoid endobronchial tumors. Atypical carcinoids are managed with lobectomy, bilobectomy or pneumonectomy with lymph node dissection [30,34,36]. Surgery, along with adjuvant chemotherapy, can be attempted for T1–T2-stage SCLC and LCNEC, but most tumors are unresectable with extensive local or distant metastasis. The mainstay of treatment includes external thoracic radiation along with etoposide and platinum-based chemotherapy [31,34,37]. Prophylactic cranial irradiation is indicated in SCLC [31,34]. Targeted biotherapy in the form of interferon-α, interferon-γ and human leukocyte interferon is being evaluated for the treatment of these lesions [34].

## 5. Gastric Carcinoids: Gastric NETs Account for Approximately 7% of All Carcinoids and 1.8% of All Gastric Neoplasms [28] and Are further Divided into Four Subtypes

Type I gastric NETs are the commonest tumors that are seen in association with chronic atrophic gastritis. These lesions often follow an indolent course. They are multifocal, small (<1 cm) tumors found in the fundus and body of the stomach. Type I gastric NETs are asymptomatic and do not warrant aggressive treatment [1,28].

Type II gastric NETs occur in patients with multiple endocrine neoplasia I, and Zollinger–Ellison syndrome. Type II tumors arise in response to the loss of tumor suppressor gene MEN 1, present on chromosome 11q13. Patients frequently complain of heartburn, peptic ulcers and diarrhea. Typically, the tumor regresses with the treatment of the underlying hypergastrinemia [28,29].

Type III gastric NETs are rare. Most lesions are solitary and greater than 2 cm in size. Type III tumors are not associated with hypergastrinemia and follow a more aggressive course than types I and II, with about 70% of tumors metastasizing to the lymph nodes [28,29].

Type IV lesions are poorly differentiated and, most often, treated with systemic chemotherapy [38].

### 5.1. Duodenal NETs

Duodenal NENs (dNETs) are heterogeneous neoplasms that are often anatomically located in the first and second parts of the duodenum, commonly at the duodenal bulb and descending duodenum. Most duodenal NETs are solitary and sporadic [39]. They are further divided into five subtypes: duodenal gastrinoma, duodenal somatostatinoma, non-functional duodenal NET, duodenal paraganglioma and poorly differentiated duodenal NEC [40]. Usually small in size and restricted to the mucosa and submucosa, duodenal NENs can metastasize to the regional lymph nodes in up to 40–60% of cases. Most cases predominantly are low-grade–well differentiated (50–70%), whereas a minority of cases (<3%) can be high-grade–poorly differentiated [41]. Duodenal gastrinomas are associated with MEN 1 and may present with Zollinger–Ellison syndrome, whereas duodenal somatostatinomas can be associated with neurofibromatosis 1 and are most often found around the ampulla of Vater [41,42]. Patients often present with various symptoms, such as abdominal pain, diarrhea and upper or lower gastrointestinal bleeding. Tumors located in the periampullary region can also lead to jaundice. Another rare complication can be partial to complete duodenal obstruction [42,43]. Most lesions are detected by upper GI endoscopy, and diagnosis is confirmed by endoscopic ultrasound and cytology. Due to the heterogeneity of duodenal NETs and the high propensity for metastasis (54.6% of cases), it is advisable to carry out early nuclear imaging as well as long-term follow-up following treatment [44]. Small submucosal tumors with no lymphovascular invasion can be treated adequately with local endoscopic excision. Endoscopic mucosal resection (EMR) for duodenal NENs between 1.0 cm and 2.0 cm in size may be insufficient, necessitating surgical resection. Full-thickness surgical excision and adequate lymphadenectomy are required for duodenal NETs greater than 2.0 cm [45].

### 5.2. Appendiceal NETs

Appendiceal NENs comprise enterochromaffin cells that secrete serotonin. Their peak incidence can be observed in the fourth decade of life with a male: female ratio of 1:2. Around 60–75% of the tumors are detected at the tip of the appendix [46,47,48]. Most appendiceal carcinoids are asymptomatic and are diagnosed post-appendectomy during histopathological examination. But, a few patients can present with features of appendicitis caused because of luminal obstruction by the tumor [46].

Immunohistochemical analysis using chromogranin A and synaptophysin can aid in confirming the diagnosis. Less than 5% of patients present with carcinoid syndrome, which depends on the amount of liver metastasis. Appendix neuroendocrine tumors can still be classified as per the histopathology-based system: Tumors with a Ki-67 index of 2% and a mitotic index (mitoses/10 high-power fields (HPFs)) of 2 are categorized as G1 and have a modest rate of proliferation. G2 tumors have a Ki-67 index between 3% and 20% and a mitotic index (mitoses/10 HPF) between 2 and 20. Finally, tumors with a Ki-67 index of 20% and a mitotic index (mitoses/10 HPF) of 20 are categorized as G3. The latter two have a greater proliferation rate; their prognosis depends on the tumor size, with a size greater than 2 cm being the most important prognostic marker. Other important prognostic factors include mesoappendiceal infiltration, the presence of vascular invasion, unclear tumor margins, metastases and concurrent functioning carcinoid syndrome. Factors like non-functionality, non-angioinvasion, Ki-67 (<2%) and mitoses < 2 cells per HPF × 40 can render a tumor to be benign and, thus, better prognostically. For management, an appendectomy is sufficient for tumors < 1 cm in size, whereas a right hemicolectomy is required for lesions greater than 2 cm in size [46,48].

### 5.3. Rectal NETs

Rectal NETs are small submucosal tumors that, fortunately, are often diagnosed early, incidental to endoscopic screening procedures. They appear as smooth, round nodules under a normal-appearing, yellowish mucosa [49,50]. These lesions are located 4–20 cm away from the dentate line and are more common along the anterior and lateral walls of the rectum [51]. Patients may present with symptoms such as a change in bowel habits, hematochezia, abdominal pain and pruritis [49,50]. As these tumors arise from non-argyrophilic cells, carcinoid syndrome is rare [50]. The rate of metastasis depends on the tumor size, muscular invasion and histopathological type. Tumors frequently metastasize to the liver, bones and lymph nodes [49,51]. Endoscopic ultrasound can be used to gauge the tumor size and the extent of invasion [49]. Tumors < 1 cm in size can be adequately treated with local excision by EMR or endoscopic submucosal dissection (ESD) [50,52]. In contrast, carcinoids greater than 2 cm in size have a greater chance of metastasizing and are treated with a more radical approach, such as by anterior resection or abdominal pelvic resection [49,50,53]. Recurrence, although rare, can occur and requires a follow-up endoscopy, rectal magnetic resonance imaging (MRI) and somatostatin receptor scintigraphy for diagnosis, followed by a planned approach for excision [53]. A recent study by Nesti et al. concluded that after complete resection of an appendiceal NET of 1–2 cm by appendectomy, right-sided hemicolectomy is not indicated. The authors underlined the clinical irrelevancy of regional lymph node metastases from appendiceal NETs [54].

## 6. Genitourinary NEN

Genitourinary NENs are a heterogeneous group of rare malignancies that can be classified as well-differentiated NETs (carcinoids), poorly differentiated NECs, which include- small cell NEC and large cell NEC, and paragangliomas [55]. These lesions arise from the amine precursor uptake and decarboxylation (APUD) cells found along the basement membrane of the genitourinary epithelium or are derived from the multipotent stem cells found in these organs. Paragangliomas usually originate from non-APUD cells [56,57]. These malignancies are more frequently seen in female GU tracts rather than in males. The most common sites are the cervix in females and the prostate in males [4,58]. Cervical NENs are often associated with human papillomavirus type 18 infection. These tumors have an aggressive course with extensive metastasis often present at the time of diagnosis. Patients present with vaginal spotting or bleeding, similar to other cervical malignancies [58,59,60,61]. The ovaries are the second most common site for NENs in females. Primary ovarian carcinoids are found within dermoid cysts. They are small, unilateral lesions that appear as yellow nodules and can give rise to carcinoid syndrome without metastasizing to the liver. Ovarian small cell NEC is a rare entity (1–2% of all ovarian cancers) that is further classified as ovarian small cell carcinoma (SmCC)-hypercalcemic type and ovarian SmCC-pulmonary type [58,60]. Endometrial NECs present in peri or premenopausal women with abnormal uterine bleeding or rarely as paraneoplastic syndromes, like Cushing’s syndrome [58,60].

Prostate carcinoids are extremely rare and account for 1–5% of all prostate cancers. Patients present with urinary complaints, often involving frequency, urgency, hematuria and dysuria. These lesions commonly arise after androgen deprivation therapy is used for prostate adenocarcinoma [55]. Testicular NENs account for <1% of testicular malignancies. In this regard, patients often complain of a testicular mass with or without pain [60]. Bladder tumors present with hematuria or urinary obstruction if the tumor is located in the bladder neck or urethra. Bladder SCNEC can present with hypercalcemia of malignancy due to parathyroid hormone (PTH)-like protein secretion [56]. Renal NENs can lead to back/flank pain and hematuria and are often found in association with horseshoe-shaped kidneys [56,57]. For resectable tumors, the treatment includes oncological resection with neoadjuvant chemotherapy. On the other hand, unresectable tumors are treated with radiotherapy and platinum-based chemotherapy regimens. Genitourinary NENs are complex and aggressive tumors and require a multimodality approach to treatment [56,57,60,61].

## 7. Pancreatic NENs

In the pancreas, the islets of Langerhans contain approximately 50% beta cells, which produce insulin, 30% alpha cells generating glucagon, 10% delta cells that provide somatostatin, 3–5% gamma cells that create pancreatic polypeptide and less than 1% epsilon cells that produce ghrelin [62]. The pancreas also contains gastrin-secreting G cells and vasoactive intestinal peptide (VIP)-secreting D2 cells [63,64]. When these hormone-producing cells become cancerous, they are referred to as pancreatic NENs (PNETs) [65]. The presentation of PNETs can vary based on the type of cell involved (Figure 3). PNETs can be divided into two categories: functioning, which causes hypersecretion syndromes by generating excess hormones, and non-functioning, which either does not create hormones or produce them at a level that does not cause clinical symptoms [62,66]. Most PNETs are non-functioning [62,66]. The World Health Organization’s (WHO) 2017 classification separates PNETs into well-differentiated NENs of the pancreas (panNET), poorly differentiated pancreatic NECs (panNEC) and mixed neuroendocrine–non-neuroendocrine neoplasms. Well-differentiated panNETs are further divided into low-grade (G1), intermediate-grade (G2) and high-grade (G3) categories, with significant changes in the classification. The updates include the inclusion of a newer subcategory of “well-differentiated high-grade NET (G3)”, changes in the Ki-67 cutoff for panNET G1, changes in terminology used for mixed neoplasms and recommendations for Ki-67 evaluation and reporting [66,67,68].

The recent classification has adopted the separation of well-differentiated NET G3 from NEC. The pancreatic NEC (panNEC; pancreatic neuroendocrine neoplasm NEN-G3) is classified into neuroendocrine tumor-G3 and NEC-G3. The distinction is clinically significant since the two respond differently to chemotherapy. Evidence is mounting on NET-G3 chemical characteristics, which supports the notion that NET-G3 is more closely related to well-differentiated NET. Several molecular indicators, such as a high Ki-67 labeling index, loss of retinoblastoma protein (Rb) expression, or KRAS (Kirsten rat sarcoma virus) mutation, can help predict the efficacy of platinum-based chemotherapies [69,70]. Advances in imaging modalities are now available for PNET screening. The imaging findings for panNEC-G3 are very different from those for panNET. On CT and endoscopic ultrasound (EUS), panNET typically appears as a clearly demarcated, internally homogeneous, hypervascular tumor, whereas panNEC-G3 shows a hypovascular pattern [70]. A significant challenge in managing PNETs is determining their prognosis, with some studies suggesting a change in current recommendations and consideration of the anatomical location as a prognostic factor [65,67,71,72]. Due to complex pancreatic anatomy, multiple pancreatic neoplastic lesions have had delayed treatments. But, of late, endoscopic ultrasound-guided radiofrequency ablation (EUS-RFA) has evolved, particularly for panNETs. This has enabled us to precisely place needles to preferentially target lesions. Armellini et al. demonstrated that this modality is equivalent and can be used instead of surgery for low-grade functioning and non-functioning panNETs [73].

### 7.1. Insulinomas

Insulinomas are tumors of the pancreas that can be either benign or malignant and, in rare cases, they may be part of the multiple endocrine neoplasia type 1 (MEN1) syndrome [74]. The primary role of insulin is to regulate blood glucose levels, but in insulinomas, the hormone is secreted inappropriately and intermittently, causing episodic hypoglycemic symptoms [62,75]. These symptoms are diagnosed based on the presence of “Whipple’s triad”, which includes hypoglycemia, low serum glucose and improvement in symptoms with glucose treatment [76]. Hypoglycemia symptoms can be divided into two categories: neurogenic and neuroglycopenic. Neurogenic symptoms, such as tremulousness, palpitations and diaphoresis, result from sympathoadrenal involvement and can be either adrenergic or cholinergic in nature [62]. Neuroglycopenic symptoms, on the other hand, are the central nervous system’s response to reduced glucose availability and may include syncope, confusion, vision changes, anxiety, convulsions, coma, amnesia and, rarely, psychiatric symptoms [62,77]. To diagnose insulinomas, six criteria must be met: blood glucose levels must be less than or equal to 40 mg/dL, insulin levels must be greater than or equal to 36 pmol/L, C-peptide levels must be greater than or equal to 200 pmol/L, proinsulin levels must be greater than or equal to 5 pmol/L, beta-hydroxybutyrate levels must be less than or equal to 2.7 mmol/L and there must be no sulfonylurea metabolites in plasma or urine [78]. Ruling out other causes of hypoglycemia is also important in the diagnostic process. The 72 h fasting test is considered the “gold standard” for confirming the diagnosis [78], although some studies suggest that a 2 h oral glucose tolerance test may be an alternative in an outpatient setting [79]. Most patients develop symptoms within 48 h [80]. Although pharmacotherapy is available, complete surgical excision is the preferred curative treatment, and incomplete resection may lead to persistent symptoms [78,81].

In patients with “hypoglycemic unawareness” caused by repeated episodes of hypoglycemia, a continuous glucose monitoring system may be useful for diagnosis. Advances in treatment include endoscopic-ultrasound-guided radiofrequency ablation for benign insulinomas [82].

### 7.2. Glucagonomas

Glucagonomas represent less than 10% of PNETs [62]. Some cases of glucagonoma are associated with MEN 1 syndrome (sometimes termed Wermer’s syndrome). An uninhibited glucagon secretion results in an increase in blood glucose levels due to an increase in hepatic glucose production and an increased breakdown of fatty acids [83,84]. The diagnosis of glucagonoma is challenging because cutaneous lesions may be the first and only evident symptom. A characteristic dermatosis associated with glucagonoma is called “necrolytic migratory erythema” (NME) [85]. NME is also associated with celiac disease, cirrhosis and pancreatitis and can occur in cases referred to as “pseudo glucagonoma”, which presents with NME but not glucagonoma, adding to the difficulty of diagnosis [62,86]. Glucagonoma with NME is defined as glucagonoma syndrome (GS) [87]. NME commonly occurs in the fingers, lower limbs, perioral region, trunk, groin, intergluteal region and genital area [83]. Typical lesions appear as pruritic, irregular erythematous lesions first, with subsequent necrosis and crusting of the central part causing bullae, which lead to ulceration, crusting, scaling and healing superimposed with hyperpigmentation. An atypical rash can also be observed in some patients [88]. The rash has a waxing and waning nature and can be confused with conditions that include intertrigo, contact dermatitis, acrodermatitis enteropathica, zinc deficiency and many other skin manifestations [83,85]. An increase in glucagon can also cause systemic symptoms, such as weight loss, diarrhea, angular stomatitis, cheilitis, diabetes mellitus, DVT, anemia and neuropsychiatric conditions [83]. Glucagon secretion can be episodic. A serum glucagon level of more than 500 to 1000 pg/mL (normal 50 to 150 pg/mL) can diagnose glucagonoma [62]. Pharmacotherapy is for patients who have advanced glucagonoma and who are not candidates for operation. Surgical resection is the only curative treatment. Other optional operations are also available and include simple enucleation, distal pancreatectomy with splenectomy, spleen-preserving distal pancreatectomy and others [89].

### 7.3. Gastrinoma

Gastrinomas are the most common functioning PNETs or gastroenteropancreatic NENs (GEP-NETs) associated with MEN1 syndrome [62]. Gastrinomas are characterized by the excessive secretion of gastric acid due to a secretagogue produced by the neuroendocrine neoplasm, which can be located in the pancreaticoduodenal or non-pancreaticoduodenal regions. This secretagogue has a similar structure to human antral gastrin, and the associated syndrome is referred to as Zollinger–Ellison syndrome, which encompasses a triad of clinical findings, including peptic ulceration in the jejunum, gastric acid hypersecretion and an islet cell tumor in the pancreas [90,91,92].

Hyperchlorhydria resulting from gastric acid hypersecretion causes symptoms such as diarrhea and acid-related peptic disease, as well as non-specific symptoms, such as weight loss, abdominal pain and heartburn. Zollinger–Ellison-syndrome-related ulcers can occur in unusual locations, such as in the third part of the duodenum and small intestine, and can lead to strictures, bleeding, penetration and perforation. While chronic gastroesophageal reflux disease with heartburn is a manifestation of Zollinger–Ellison syndrome, such patients more frequently experience esophageal strictures caused by acid reflux [93].

The early stages of Zollinger–Ellison syndrome are often missed due to the widespread use of proton pump inhibitors (PPIs), which normalize gastric acid hypersecretion, and the disease is rarely suspected [94]. To diagnose gastrinoma, a fasting serum gastrin level of 1000 pg/mL can be used after excluding achlorhydria and discontinuing all additional medications, including antacids, for at least one week. In patients with intermediate gastrin levels (100 to 1000 pg/mL), a secretin test can be performed. During the test, secretin normally has an inhibitory effect on gastrin release; however, in gastrinoma patients, it will increase gastrin release. A rise in gastrin levels of more than 120 pg/mL after secretin administration of 2 U/kg is 100% specific for Zollinger–Ellison syndrome [93].

For patients with negative secretin test results but a high clinical suspicion for Zollinger–Ellison syndrome, a calcium stimulation test with calcium gluconate can be performed. Calcium-sensing receptors (CaRs) are expressed on gastrinomas and mediate calcium-stimulated gastrin release. A rise in gastrin levels of more than 50% from baseline is considered positive [62,95,96]. To exclude MEN1, fasting calcium, procalcitonin and serum prolactin levels should be investigated.

The pharmacotherapy for gastrinomas includes histamine receptor antagonists, PPIs and somatostatin analogs (SSAs). Some preclinical and clinical studies have also reported tumor-suppressive effects on well-differentiated NETs [94,97]. Most patients with advanced disease present with diffuse liver metastases, and surgical resection is only possible in a few cases. Patients with progressive, non-resectable metastatic disease require treatment with antitumor non-surgical approaches [98].

### 7.4. Somatostatinoma

Somatostatinomas mainly originate from two organs: δ cells of the pancreas that produce somatostatin and from the duodenum [99]. These tumors infrequently occur as part of MEN 1 syndrome (MEN 1 also known as Wermer syndrome—involves tumors of the pituitary gland, the islet cells of the pancreas and the parathyroid gland). Duodenal somatostatinoma also occurs with von Hippel–Lindau syndrome, neurofibromatosis type 1 and tuberous sclerosis [97,98]. Somatostatin has an inhibitory effect on endocrine and exocrine secretory functions. The somatostatinoma syndrome comprises weight loss and abdominal pain, and less often comprises diabetes mellitus, cholelithiasis and diarrhea/steatorrhea. This syndrome mostly occurs in somatostatinomas that are found localized in the pancreas [100]. The duodenal location of the somatostatinoma causes anemia and gastrointestinal hemorrhage [99]. A fasting plasma hormone concentration of more than three times the normal concentration can be present. In the case of an intermediate result (not more than three times normal), stimulatory tests, including secretin or calcium stimulation tests, can be used [62]. It is helpful to keep the differentials in mind as somatostatin levels are also increased in medullary carcinoma of the thyroid, lung cancers, pheochromocytomas and paragangliomas. Since their presentation is generally late, tumors at diagnosis are easily visualized by imaging modalities. Patients need nutrition support or hyperalimentation. Treatment with long-acting somatostatin analogs has been successful in a few cases. Excision of the primary tumor and the metastasized lymph nodes remains the only curative treatment. Tumors with metastases (liver and lymph node) receive treatment with tumor debulking surgeries and liver resection to cause symptom relief resulting from mechanical obstruction and heavy load of tumor [101].

### 7.5. VIPoma

VIPoma tumors are commonly seen in the pancreas, but also along the sympathetic chain, for example, in ganglioneuromas, ganglioneuroblastomas or neuroblastomas, as VIP functions as a neurotransmitter. Pheochromocytomas secreting VIP have also been described [102]. Similar to other PNETs, VIPoma occurs as a part of MEN1 [62]. In the gastrointestinal tract, VIP has the following actions: it enhances the contraction of enteric smooth muscle cells, augments secretion from the exocrine pancreas, increases gastrointestinal blood flow and inhibits gastric acid secretion [102]. Patients present with a classic syndrome called “WDHA syndrome” also called “Verner–Morrison syndrome” characterized by diarrhea that is watery with about 20 bowel movements with a daily volume of stool exceeding 3 L, hypokalemia and achlorhydria [103,104]. The key feature is large-volume secretory diarrhea due to which higher amounts of bicarbonate and potassium are lost in the stool, resulting in hypokalemia, metabolic acidosis and the depletion of volume. Hypochlorhydria and achlorhydria are evident, but not always. Other reported symptoms include hypocalcemia, flushing and glucose intolerance, among others [103]. Patients develop complications related to diarrhea, such as electrolyte abnormalities and renal failure, and warrant resuscitation with intravenous fluids and electrolytes. The mainstay of management for such patients is symptomatic with somatostatin and its analogs [105]. A diagnosis of VIPoma is, hence, suspected when there is the presence of secretory diarrhea and VIP levels more than 75 pg/dL, but a firmer diagnosis can be made when elevations (>200 pg/dL) are present [104]. Surgical resection is used for localized and resectable forms that metastasize, but seldom achieve a definitive cure. Metastatic VIPomas present the two-way challenge of managing the tumor burden and secretory syndrome; therefore, sunitinib and chemotherapy are the two therapeutic options that combine antitumor and antisecretory efficacies effectively for treatment [106].

### 7.6. Other Pancreatic NENs

Besides the PNENs described above, the other functional hormone-secreting PNENs include “GRFomas”, which generate growth hormone releasing factor (GRF); “ACTHomas”, which secrete ACTH; “PTHrPomas”, which produce parathyroid hormone-related protein (PTHrP); and “PNETs causing carcinoid syndrome”, which secrete serotonin and tachykinins [107]. There are also rare cases of PNETs secreting hormones such as renin, erythropoietin, insulin-like growth factor 2 (IGF-2) or luteinizing hormone [108]. Tumors without related clinical features of hormone overproduction, also called non-functioning, present with jaundice, abdominal pain, weight loss, or other non-specific symptoms or are diagnosed incidentally [109].

## 8. Adrenocortical Tumors

Adrenocortical tumors include primary bilateral micronodular or macronodular disease, adenomas, and carcinomas. Due to their various causes, clinical presentations and eventual outcomes, these neoplasms must be appropriately classified. Rarely seen are primary retroperitoneal NENs. Adrenal cancers include adrenal cortex carcinoma, malignant pheochromocytoma, malignant lymphoma and neuroblastoma. Adrenal NEN, or pheochromocytoma of the adrenal medulla, is a well-known form of tumor that is usually harmless but can sometimes progress to malignancy [110,111].

Pathologists have many important responsibilities when evaluating adrenocortical lesions, including confirming their adrenocortical origin, diagnosing malignancy, providing relevant prognostic information on adrenocortical carcinoma and correlating laboratory results with clinicopathologic findings. Rare and deadly, adrenocortical carcinoma is a type of endocrine malignancy. Diagnostic difficulties with these malignancies have led to the ongoing proposal of diagnostic algorithms and criteria. Important distinctions from other tumor types include myxoid and oncocytic variations. Oncocytic adrenal carcinomas also have distinct diagnostic criteria compared to other types of carcinomas. Although rare in children, adrenocortical carcinomas can develop at any age. The histologic characteristics of malignancy present in an adult tumor may not be associated with aggressive disease in a youngster, making diagnosis difficult. Beckwith–Wiedemann and Li–Fraumeni syndromes are associated with an uptick in the incidence of adrenocortical carcinomas, but the vast majority of cases still arise independently. Adrenocortical carcinomas and adenomas can be distinguished from one another, and carcinomas can be divided into prognostic categories using gene expression profiling by transcriptome analysis [110].

Recent research has shown that positron emission tomography (PET)-CT is an effective diagnostic tool for determining whether adrenal lesions are malignant or benign. Sensitivity for dynamic CT ranges from 61% to 100%, while specificity for MRI ranges between 82% and 97%. Furthermore, any metastases detected on a dynamic CT scan with a washout percentage of less than 37–50% or an absolute washout percentage of less than 60% are considered suspicious. The formula for determining the percentage of washout is as follows: washout % = 100 ([EA DA]/EA), where EA is the attenuation (in Hounsfield units) on contrast-enhanced scans and DA is the attenuation on delayed contrast-enhanced images (usually at 10 min after starting injection). The following formula is for absolute percentage washout: 100 × ([EA DA]/[EA attenuation value at unenhanced CT]) [110,111]. If the adrenal tumor is 4 cm or larger, surgical removal may be necessary because of the high malignancy risk. The “gold standard” for the surgical removal of benign adrenal neoplasms, laparoscopic adrenalectomy, is now also commonly utilized for malignant tumors. If the patient’s overall health is stable after surgery or if the tumor returns, chemotherapy with drugs like cisplatin and etoposide may be an option. In the adrenal gland, however, questions remain as to whether severe surgery may be necessary and, if so, whether or not a particular chemotherapy treatment should be administered. A higher rate of recurrence, peritoneal carcinomatosis, positive margins and local recurrence has been documented after laparoscopic surgery, as compared to open surgery in several retrospective series [110,112,113].

## 9. Other Rare NENs

Head and neck NENs can be of epithelial origin or neuronal origin (e.g., paragangliomas and olfactory neuroblastomas). Notably, they are present in the larynx but are also seen in other locations that are nasal, parotid, hypopharynx and tongue. Based on site, patients present with dysphagia, hoarseness, epistaxis, facial mass and neck mass [114].

Thymus NENs are aggressive mediastinal tumors that do not show symptoms in only 30% of cases and are discovered incidentally after a routine radiograph of the chest. More commonly, they present with non-specific symptoms (cough, chest pain, shortness of breath, weight loss, asthenia and chronic fever) or present with SVC syndrome [115].

Thyroid NENs, like medullary carcinoma of the thyroid, have their origin in C cells of the thyroid gland that produce calcitonin. They present as thyroid nodules or symptoms associated with MEN [116].

Breast NENs are mostly seen in postmenopausal women presenting as a single breast nodule that, in some cases, is associated with other local signs (painful axillary adenopathy, bloody nipple discharge and nipple retraction) [117].

Skin NENs, like Merkel cell carcinoma, affect the skin of the head and neck region in older individuals. They present with wart-like, blister-like or black-eyed-pea-shaped lesions, usually without ulceration [118].

## 10. Genetics and Syndromes

From a genetic standpoint, NECs have inactivation of *RB1* and *TP53* that, in contrast, occur rarely in NETs [119]. Hereditary NET syndromes present in common patterns, where mainly two pathways are involved: the regulation of the cyclin-dependent cell cycle (especially in MEN1 and MEN4) and the involvement of the PI3K/mTOR pathway (especially in autosomal dominant syndrome caused by a mutation in the tumor suppressor gene on the 11q13 chromosome encoding the menin protein, characterized by primary hyperparathyroidism, duodenopancreatic neuroendocrine neoplasms (NETs) and anterior pituitary tumors [120,121].

### 10.1. Multiple Endocrine Neoplasia 2 (MEN 2)

An autosomal dominant disorder as a result of mutation of the RET proto-oncogene on chromosome number 10, it has two different variants that are known as MEN 2A and MEN 2B. MEN 2A and MEN 2B have the risk of medullary carcinoma of the thyroid and pheochromocytoma. In addition, MEN2A has a risk of primary hyperparathyroidism, and MEN 2B presents with marfanoid body habitus [120,122].

### 10.2. Multiple Endocrine Neoplasia 4 (MEN 4)

An autosomal dominant disorder deriving from mutation of a cyclin-dependent kinase inhibitor (CDKN1B) on chromosome number 12 coding for the p27 protein. Pituitary tumors and hyperparathyroidism are commonly seen in MEN 4, but gastrointestinal and pancreatic tumors are less commonly seen than in MEN 1 [120].

### 10.3. Von Hippel–Lindau Syndrome

An autosomal dominant syndrome resulting from mutation of the VHL gene on the 3p25 chromosome. Its features are retinal angiomas, hemangioblastoma of the cerebellum and spinal cord, endolymphatic sac tumors, pheochromocytoma, renal cell carcinoma and pancreatic cystic lesions including serous cystadenomas and NETs [120].

### 10.4. Neurofibromatosis 1 (NF 1)

An autosomal dominant disease arising from the mutation of NF1 on chromosome 17q11, NF1 presents with spinal malformations, vascular malformations and malignant and benign tumors in the peripheral and central nervous systems with 2% of patients also having neuroendocrine tumors [120,123].

### 10.5. Tuberous Sclerosis

An autosomal dominant disease stemming from mutation of tuberous sclerosis complex (TSC) 1 on the 9p34 chromosome or TSC2 on16p13.3 chromosome, which are genes that encode for the hamartin and tuberin proteins, respectively, tuberous sclerosis has the classic development of hamartomas in almost every organ, and patients complain of disabling neurologic features, dermatologic features and hamartomatous lesions that are tumor-like (cortical tubers), lymphangiomyomastosis, cardiac rhabdomyoma, subependymal nodules and renal angiomyolipomas. Both functioning and non-functioning are seen in patients with TSC [120,124].

### 10.6. Pheochromocytoma and Paraganglioma

Paragangliomas are uncommon neuroendocrine tumors that secrete catecholamine (norepinephrine) and are typically found in the pre-aortic and paravertebral sympathetic plexus or cranium base. The less differentiated tumors comprise head-and-neck paragangliomas in the jugular foramen, ear or carotid body. They secrete norepinephrine in contrast to adrenal medulla tumors, which primarily secrete epinephrine. The effects of epinephrine and norepinephrine on adrenergic receptors manifest primarily as migraines, palpitations and excessive perspiration. Catecholamine-producing tumors resemble paroxysmal conditions with hypertension and/or cardiac rhythm disorders, such as panic attacks. These tumors may be sporadic or the result of one of a number of genetic conditions [125].

## 11. Diagnosis

NENs are typically diagnosed based on their clinical presentation, signs and symptoms, with diagnostic parameters such as serum markers, also known as tumor markers, used in conjunction with or in addition to imaging modalities and immunohistochemistry [126].

### 11.1. Imaging

Imaging modalities, including CT, MRI and ultrasound, can be used in the diagnosis of NETs. Cross-sectional radiography is crucial for determining where the main tumor is located and for spotting metastases. Although MRI detection rates are now competing with CTs as the primary modality due to technical advancements, CT remains the most common type of imaging [127].

As nuclear or molecular imaging techniques are thought to have a complementary role in the localization and staging of NENs, they are used in conjunction with morphologic (given by conventional radiology) and functional (given by nuclear/molecular imagery) techniques. Routinely, conventional imaging is used for localization, but its sensitivity is suboptimal due to the small size and widespread distribution of lesions. Functional imaging with indium-111-octreotide detects increased expression of somatostatin receptors by tumors with greater sensitivity than conventional imaging. For PET/CT imaging, positron-emitting radionuclide-labeled peptides with increased detection rates have been developed. Gallium-68-peptide PET/CT is the new imaging “gold standard” for NETs, with a sensitivity and specificity exceeding 90% [128]. When using Ga-68 DOTA-peptides and F18-FDG in patients with neuroendocrine tumors, both PET and CT should be carried out. Ga-68 DOTA-peptide uptake is increased in low-grade NETs, whereas F18-FDG PET/CT uptake is elevated in G2 and G3 NETs. In cases where Ga-68 DOTA-peptide PET/CT is negative or marginally positive, F18-FDG should be administered to determine the status of dedifferentiation. The 68Ga-labeled analog of somatostatin PET/CT has emerged as a crucial imaging modality for NEN. It has the potential to be the first-line imaging investigation for evaluating these tumors, particularly GEP-NETs, and can influence numerous facets of their management. Gallium-68-labeled fibroblast activation protein inhibitor PET/CT may perform better than F18-FDG PET/CT in staging lung cancer, notably in detecting metastases to the brain, bone, lymph nodes and pleura. However, its role is limited in the case of somatostatin-negative NENs. Such investigations not only aid in diagnosis but also can help direct the next step of therapy, to cite an example: FDG PET/CT can be utilized to plan further management, including peptide receptor radionuclide therapy (PRRT) [117,118].

In the case of insulinomas, the visualization of insulinomas can be achieved using a variety of conventional screening methods, including EUS, MRI, CT, transabdominal ultrasonography, somatostatin receptor imaging (111In-octreotide scintigraphy or 68Ga-DOTATOC), PET/CT (68Ga-DOTATOC PET/CT), 18F-FDG PET/CT, intraoperative ultrasonography and arterial calcium stimulation with hepatic venous sampling [129]. PET/CT with 68Gallium-exendin-4 has been shown to be useful in diagnosing occult insulinomas [88,130].

#### 11.1.1. CT

For quite a long time, CT has been the primary screening approach employed for NEN identification, grading, judgment, and treatment response assessment [128,131]. The relatively low rate of bone metastases and small (1 cm) infiltrating lymph nodes are potential drawbacks of this type of imaging. It has been observed that overall NEN detection has a mean of 82% and 86% precision and accuracy, with higher rates for pancreatic and hepatic pathology [128,132,133]. The radiation dose given to patients who are receiving long-term follow-up with imaging surveillance is a potential drawback of CT imaging and varies depending on the CT scanner type and examination methodology [134].

#### 11.1.2. MRI

When compared to CT, MRI offers superior picture contrast, and the utilization of multiple MRI sequences further improves diagnostic accuracy [128]. For detecting potential liver and pancreatic tumors as well as spotting metastatic disease in the bones and brain, MRI has been proven to be superior to CT [34,128,131]. Lately, it has become apparent that the degree of hepatic involvement—measured as the percentage of hepatic tissue replaced by tumoral tissue—is significantly prognostic and influences the choice of a particular therapeutic approach, especially when it comes to cytoreduction via surgical or ablative methods [131]. The most effective method for detecting tiny hepatic metastases from NENs is to use diffusion-weighted MRI sections or IV contrast [128].

#### 11.1.3. Ultrasonography and Related Applications

Conventional abdominal ultrasonography has a generally low detection rate for PanNENs at about 40% and is regarded as an operator-dependent imaging method. It performs better in detecting hepatic metastases [128]. In contrast, EUS, which also depends on the operator, offers a more sensitive method for identifying PanNENs, with a mean detection rate of 92% (range, 74–96%) [135]. Lesions at the pancreatic tail have reduced detection rates, but duodenal neoplasms and nearby lymph nodes have reported detection rates of about 63% [136]. Furthermore, EUS permits access to tissue sampling, which enables verification of the diagnosis while gaining grading information [136].

### 11.2. Tumor Markers

Serum chromogranin A (CgA) and neuron-specific enolase (NSE) measurements have traditionally been considered as tumor markers for NENs. However, the poor performance of these markers in terms of their sensitivity and specificity has been well documented [8]. Despite this, these markers remain widely used in clinical practice consequent to their availability and ease of use.

Recently, a new test known as the “NETest” has been extensively studied for its potential use in the diagnosis of NETs. The NETest is a blood test that measures the levels of five different markers (CgA, NSE, IGF-2, pancreastatin and glycoprotein hormones) associated with NETs [137]. The NETest has been shown to have higher sensitivity and specificity, as compared to traditional markers (like CgA and NSE), and holds promise as a valuable diagnostic tool in the future [137]. Further studies are needed to fully establish the role of the NETest in the diagnosis of NETs and its integration into clinical practice [137].

In conclusion, the diagnosis of neuroendocrine tumors is a complex process that involves multiple factors and tests, including both biochemistry and imaging studies. The use of specific tumor markers, like the NETest, along with other relevant clinical and laboratory data, can greatly aid in the diagnosis and management of these neoplasms.

#### 11.2.1. Specific Tumor Markers

NET cells are known to lack fine processing of biologically induced peptide hormone synthesis [138]. The measurement of these peptides and their progenitors, such as amino acids, can assist in the diagnosis of NETs (Figure 4) [138]. In some cases, this measurement can also provide information about the size of the tumor [139]. In instances where the production of multiple hormones is evident, these levels may change as the condition progresses [139]. The measurement of serum hormone levels can be useful in detecting non-functioning tumors, where there is no relationship between the clinical symptoms and hormone products [138,140]. Recent research has revealed that the two subunits of human chorionic gonadotrophin (*hCG*), α and β, may be indicative of medullary thyroid cancer, small cell carcinoma of the lungs and non-functioning gastroenteropancreatic neuroendocrine tumors [138,140].

#### 11.2.2. Non-Specific Tumor Markers and Immunohistochemistry (IHC)

Increasingly, NET markers are understood to include other proteins that regulate the synthesis, metabolism and release of hormones and specific hormones secreted by NE cells [138,139,141]. Three soluble acidic monomeric proteins—CgA, CgB and CgC—are found in the same granules that secrete the existing peptides [139,142]. Whereas chromogranins, including CgA-negative ones, are important, CgA is the most commonly used granin in clinical practice, especially as CgB-positive cancers become more prevalent [139].

In several NETs, such as pheochromocytomas [143,144], paragangliomas [140,145], carcinoid syndrome, the islet of Langerhans cell tumors of the pancreas [143,145,146], medullary thyroid cancer [147] and parathyroid and pituitary adenomas [148], plasma CgA levels may be high, although this is much less common (60%) in SCLC [140,149]. Metastatic carcinoids and GEP tumors have been shown to have the highest CgA levels [144,149,150,151]. When interpreting CgA data, it is crucial to consider both the tumor load and secretory activity [143,151,152]. False positive CgA test results are often caused by renal failure and hypergastrinemia [140,143]. Several assays using either monoclonal or polyclonal antibodies have been developed to measure intact CgA and its various cleavage products, with significant variations in sensitivity and specificity [152]. Until an internationally recognized standard for CgA is established, this should be taken into account [152]. Studies comparing foregut carcinoids to specific biological markers have shown that foregut carcinoids, including bronchial, thymic, head and neck primary, have a higher sensitivity to CgA, similar to a specific tumor marker’s sensitivity in patients with pheochromocytomas and ileal carcinoid syndrome [140,143,151,152]. Additionally, as CgA correlates with both tumor burden and biological activity, it is a predictive factor for midgut carcinoids [145,146].

**Figure 4 jcm-12-05138-f004:**
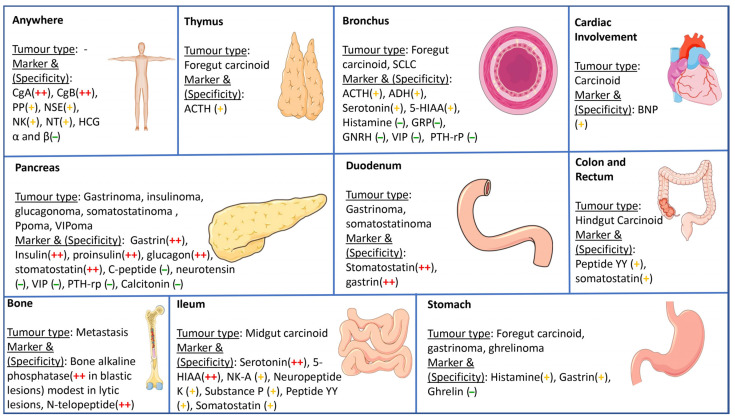
Specific biochemical markers for different NETs [153]. VIP: vasoactive intestinal peptide, NSE: neuron-specific enolase CgA: chromogranin A, CgB: chromogranin B, HCG: human chorionic gonadotropin, ACTH: adrenocorticotropic hormone, 5HIAA: 5-hydroxyindoleacetic acid, GRP: gastrin-releasing peptide, GNRH: gonadotropin-releasing hormone, PTH-rP: parathyroid hormone-related protein, peptide YY: peptide tyrosine–tyrosine, ADH: antidiuretic hormone, NK: neurokinin, NT: neurotensin, (++): high specificity-red, (+): intermediate specificity-yellow, (−): low specificity-green. *Figure credit: parts of the figure were generated by making use of pictures available from Servier Medical Art, accessed from Servier, and licensed under a Creative Commons Attribution 3.0 unported licens* Published with permission from PearResearch.

## 12. Management

The management of NENs has proven to be challenging because of their varied clinical presentations and response to therapy. The mainstay of treatment has been surgery, but with the evolution of pharmacotherapy in recent decades, the horizon of options has broadened. Most NENs express somatostatin receptors (SSTRs) leading to the first-line use of long-acting somatostatin analogs (SSAs) for symptomatic control [154]. Other treatment options include everolimus [155,156], sunitinib [156], chemotherapy, interferon-α and PRRT, which is especially efficacious in SSTR-expressing NETs [157]. Due to the variation in the biological behavior of these neoplasms, the combined use of nuclear imaging with and 18F-FDG-PET and 68Ga-DOTA-TOC/TATE/NOC-PET scans (dual-baseline scanning) prior to therapy with PRRT has been proposed in recent years. The detection of SSTR overexpression by 68Ga-DOTA-TOC/TATE/NOC-PET scans and the localization of foci with high glycolytic metabolism via 18F-FDG-PET scans can be used for personalization and the selection of patients for PRRT initiation [158,159,160]. Since the treatment, prognosis and metastatic potential differ based on the location of these tumors, treatment guidelines vary and often require a personalized approach targeting the patient and not just the disease. The possible treatment options for NETs are summarized in Figure 5.

### 12.1. Gastroenteropancreatic NETs (GEP NETs)

Types 1 and 2 Gastric NETs smaller than 1 cm (G1) without angioinvasion and not extending to the muscularis propria can benefit from endoscopic follow-up. Tumors between 1 and 2 cm in size can benefit from EMR and endoscopic polypectomy, provided that high-risk signs, like angioinvasion, higher-grade G2/G3, extension to the muscularis propria and size > 2 cm, are not present. In such cases, total gastrectomy with lymphadenectomy or antrectomy should be carried out. Types 3 and 4 can benefit from radical resection and metastatic lymph node dissection. Duodenal NETs can be managed similarly [69,161,162,163]. For medical management, SSAs are only recommended for recurrent and multiple G1 gastric and duodenal NETs [164].

For small bowel NETs, the aim of surgical treatment must be to achieve a curative radical resection. This can be achieved by bowel resection, adequate loco-regional lymphadenectomy, through the dissection of the mesentery. This ensures improved survival. In cases where the tumor is present in the terminal ileum, an additional right hemicolectomy is proposed [165,166]. For medical management, PRRT has been found to prolong progression-free survival and overall survival in SSTR-expressing small intestine NETs as a second line when first-line SSAs (octreotide and lanreotide) have failed [161,167]. Everolimus can be used as a third-line treatment if tolerated, following which chemotherapy is also a potential option [157]. Recent studies on the use of temozolomide/capecitabine combinations in the case of poorly differentiated small bowel NETs have proven to be efficacious [168,169].

As far as the management of carcinoid syndrome is concerned, the SSAs octreotide and lanreotide with dose escalation can provide substantial benefit, and recent studies have suggested them to be efficacious in refractory cases, for which the drug pasireotide is also recommended [170,171]. IFN-α can be used as an additional therapy but often may not prove well tolerated [131]. Moreover, a tryptophan hydroxylase inhibitor, telotristat, has been recommended in cases of carcinoid syndrome refractory to SSAs for symptomatic control, as studies have shown its efficacy in reducing bowel movements and facial flushing in patients [157,167]. However, the early use of PRRT, as a second line, has recently been advocated to be efficacious for use in functional pancreatic NETs that are refractory to SSAs by the European Society for Medical Oncology (ESMO) [172]. For colonic NETs sized less than 2 cm, endoscopic excision with EMR or polypectomy is recommended, whereas for NETs greater than 2 cm, poorly differentiated tumor or invasion of the muscularis propria, surgical resection via right or left hemicolectomy with adequate lymph node dissection is advised.

For rectal NETs, curative resection of the lesion is the mainstay of treatment. If the lesion is smaller than 1 cm in size, endoscopic removal alone may suffice. However, if the size is from 1 to 2 cm, with no extension to the muscularis propria, local resection can be performed. For high-grade tumors exceeding 2 cm sizes, total mesorectal excision is advised via the anterior or abdominoperineal route [173,174]. In other words, excision via endoscopic mucosal resection (EMR) in the case of tumors smaller than 1 cm can be achieved, and for lesions between 1 and 2 cm without lymphatic extension and limited to the submucosa, resection via endoscopic submucosal dissection (ESD) or transanal endoscopic microsurgery (TEM), instead of EMR, is advised. The sole local resection procedure, whether endoscopic or transanal, has proven to be efficient in ensuring a radical resection and a very low recurrence rate for T1 lesions of the rectum. Node-positive or T2 lesions should be thoroughly examined using PET/CT. Once remote metastases are ruled out, a low anterior resection (LAR), total mesorectal excision (TME) or excision through the abdominoperineal route can potentially be performed. In the case of rectal NETs extending to the muscularis, low anterior resection and intersphincteric resection may be helpful [117]. For the medical management of G1 and G2 colorectal NETs, the RADIANT-2 trial found that the combination of SSAs and everolimus helped prolong progression-free survival, and is thus advised, with SSA being the first-line [175]. Chemotherapy may be an option for advanced stages. However, further studies are required to confirm the efficacy.

For low-grade non-functioning pancreatic NETs, a conservative approach with follow-up is encouraged [176]. Whereas high-grade tumors, regardless of the size, must be surgically resected via pancreaticoduodenectomy if head presentation and distal pancreatectomy and splenectomy if body and tail presentation are seen [177,178]. However, in the case of insulinoma, instead of radical excision, parenchyma-sparing surgery or enucleation has been reported to have higher benefits and lesser complications [177]. In non-functioning pNETs associated with MEN1 syndrome, measuring less than 2 cm in size, conservative management is advised, although surgery is recommended when lesions exceed 2 cm in size [177]. Moreover, the management of MEN1 gastrinomas greater than 1 cm in size is advised to be surgical, preferably pancreaticoduodenectomy, to avoid liver metastases and achieve a better prognosis [179,180]. For functioning tumors, debulking surgery may be an option.

As far as medical management is concerned, proton pump inhibitors are the treatment of choice in Gastrinomas: SSAs are advised for antiproliferative advantage, specifically long-acting, like lanreotide and octreotide [181]. Interferon-α can be added only if well-tolerated; however, targeted therapy with everolimus and Sunitinib is advised for G1 and G2 progressive pNETs as the second line. For extension into the liver, locoregional treatment methods, like hepatic artery embolization (HAE), selective internal radiation therapy (SIRT) and radiofrequency ablation (RFA), are recommended [157]. Lastly, chemotherapy options include streptozocin (STZ), 5-fluorouracil (5-FU) or temozolomide, individually or in combination with capecitabine. Platinum-based therapy followed by FOLFOX/FOLFIRI regimens is definitively recommended for G3 NETs and progressive disease [182,183,184].

Liver metastases in GEP NETs: Liver metastases are a natural occurrence observed in NETs based on the primary tumor location. Pancreatic and small intestine NETs show 28% to 78% and 67% to 91% rates for liver metastases incidence respectively, whereas rectal, appendiceal, and gastric NETs rarely lead to liver metastases [157]. Consideration of liver surgery for such metastases is based on an assessment of multiple factors, for example, tumor grading (it is advisable that G1–G2 tumors should only be considered for liver surgery, although G3 tumors can have potentially high recurrence rates, even after surgery). Other factors include extra-hepatic involvement, normal liver remnant volume and symptoms [185]. ENETS has proposed the criteria for curative surgery only in the case of a resectable G1 or G2 liver disease with low morbidity and mortality less than 5%. Moreover, there should be no sign of right heart insufficiency, nor should there be the presence of any unresectable lymph nodes, extra-abdominal metastases or peritoneal carcinomatosis [157].

Liver metastases deriving from NETs have three morphological types, namely, type I (which is a single metastatic lesion), type II (isolated metastatic lesion along with small deposits) and type III (scattered and disseminated metastasis) [186]. Debulking surgery is not yet fully accepted but has been shown to improve quality of life in cases of failed medical management [185]. For completely unresectable disease, approaches like a two-stage hepatectomy and portal vein ligation have been proposed [187,188]. Liver transplantation may be considered in selected patients with carcinoid syndrome or extended liver metastases refractory to medical management, but it remains controversial as recent studies have shown that the extent of liver extension can be underestimated, and in up to half the cases, liver metastases from NETs are not detected on preoperative imaging. This explains, in part, the increased difficulty of achieving a curative resection as well as a higher recurrence rate post-surgically [189,190,191]. In this light, more robust evidence is required on this front.

### 12.2. Pulmonary NETs

Pulmonary neuroendocrine tumors (NETs) have different recommended treatments based on their type. Large and small cell NETs are not recommended for surgical treatment unless they are in an early, localized stage due to the risk of systemic spread. Bronchial NETs, however, can be managed with surgical resection that preserves the parenchyma and includes lymph node dissection [34]. SSAs have shown effectiveness in low-grade tumors [147], and recent trials have proven the progression-free survival-prolonging efficacy of everolimus in lung NETs [192,193]. Chemotherapy options include temozolomide alone or combined with capecitabine. For advanced disease, STZ/5-FU and cisplatin/etoposide are recommended, and PRRT has shown efficacy and may also be a viable option [194,195]. Research is ongoing for the establishment of a therapy protocol for large cell neuroendocrine carcinomas (LCNECs) with recent advancements in the identification of newer therapy targets, such as the stimulator of interferon genes (STING) pathway, which stimulates immunity towards cancer [196], mutations in the epithelial growth factor receptor (EGFR), Phosphoinositol-3 kinase (PI3K), mTOR, VEGF and human epidermal growth factor receptor 2 (HER-2) have been made; however, the upfront adoption of targeted therapy is still controversial, due to the limited available data on the efficacy [197].

A recent study on the prognostic value of surgical intervention with complete resection and systematic node dissection in LCNECs has reported a better prognosis in such patients [34]. Furthermore, given the rarity of LCNEC and the ever-evolving studies on its therapy, a robust guideline may be adopted in the near future.

### 12.3. Evolving Therapy Targets for Other NETs and Future Perspectives

The current systemic therapy for NENs mainly targets tumor proliferation and hormone production. The extensive research in this domain has allowed the identification of various drug targets that have been exploited using currently approved and under-trial molecules (Figure 6). Therapy with SSAs, like octreotide and lanreotide, has been first line due to their anti-tumor properties and well-established antiproliferative actions [198]. Interferon-α with its cytotoxic, anti-proliferative, and pro-apoptotic properties [199], along with PRRT, makes targeted radionuclide action on tumor cells possible and is a second line [200]. Everolimus, an mTOR inhibitor regulates neuroendocrine tumor cell proliferation and has been efficacious in increasing progression-free survival in various trials [44,192]. The hypervascularity of NEN makes sunitinib (an antiangiogenic option via the inhibition of the VEGF receptor) a potential therapy, with trials proving its efficacy in increasing the progression-free survival and overall survival in pancreatic NETs [193,201]. Chemotherapy is extensively studied in G3 NETs, with temozolomide, capecitabine, etoposide and platinum being the most common in practice.

As per the current guidelines, the first line of chemotherapy in G3 NETs should be platinum-based therapy with combinations like cisplatin or carboplatin/etoposide and cisplatin/irinotecan. The suggested second-line therapy can be temozolomide, oxaliplatin or irinotecan-based regimens [201]. Further studies are required to establish an improved treatment plan and sequence, although a focus on epigenetic factors and the pathways of NET development can allow a more holistic approach. Currently, ongoing registered clinical trials are aimed at understanding the efficacy of already available treatments in different combinations, as well as to offer comparisons between them, for instance, PRRT vs. temozolomide/capecitabine or everolimus or sunitinib, etc. Furthermore, sequential treatments are also being compared, which will hopefully ultimately increase the quality of treatment of NETs [202] and allow personalization of therapy.

Peptide radionuclide receptor therapy (PRRT)—177Lu-Dotatate treatment—did not significantly enhance median overall survival in the recent NETTER-1 trial when compared to high-dose long-acting octreotide treatment. The difference of 11.7 months in median overall survival between treatment with 177Lu-Dotatate and high-dose long-acting octreotide alone may be regarded clinically important, despite the fact that the final overall survival did not reach statistical significance [73,203].

Various clinical trials are ongoing with the aim of expanding current options for monotherapy and combination therapy for NETs. Extensive research is being performed to determine the efficacy of 177Lu-DOTATOC (Lutathera), a novel therapeutic agent for PRRT in gastroenteropancreatic NETs (phase II Trial—NCT04276597). A further phase II trial (NCT05247905) is comparing the effects of the chemotherapy agents capecitabine and temozolomide (CAPTEM) and 177Lu-DOTATOC in pancreatic NETs. A phase III trial (NCT04919226) studying the efficacy of radio-conjugate 177Lu-Edotreotide, a Lutetium Lu-177-labeled SSA, in comparison to CAPTEM, everolimus, and FOLFOX (folinic acid, 5-FU and oxaliplatin) for gastroenteropancreatic NETs is ongoing. A phase I/II clinical trial (NCT04086485) is testing the combination of Lutathera with olaparib, a poly ADP-ribose polymerase (PARP) inhibitor for inoperable NETs of gastroenteropancreatic origin. Trials studying the efficacy of relatively newer kinase inhibitors, like surufatinib (phase II—NCT04579679) and Dovitinib (phase II—NCT01635907), are underway for treating NETs. A phase II clinical trial (NCT04412629) focusing on the drug cabozantinib for high-grade NETs is in process, whereas a further phase III trial (NCT03375320) testing its efficacy in advanced pancreatic NETs and carcinoids is ongoing. The trials currently underway may help provide a wider base for therapy, especially considering studies on the novel agent Lutathera. Chemotherapy agents like irinotecan and cisplatin are also under study for the evaluation of their efficacy in advanced NETs (phase II trials—NCT00353015 and NCT00004922). Other chemotherapy agents like thalidomide, pemetrexed and bevacizumab-temozolomide combination are in trials to study their efficacy in advanced and metastatic NETs (NCT00027638, NCT00424723 and NCT00137774, respectively). Newer agents like Epothilone B (EPO906), ONC201 and Procaspase Activating Compound-1 (PAC-1) are under trial for the evaluation of their potential therapeutic role in carcinoids and metastatic NETs (phase II—NCT00050349 and NCT03034200; phase I—NCT02355535, respectively).

A new combination of chemotherapy agents, BXCL701-Pembrolizumab, is being examined for efficacy in small cell NETs of prostate and adenocarcinoma phenotype (phase Ib/II—NCT03910660) and is hoped to provide promising results. Numerous trials underway are evaluating combinations of therapies for better neoplastic control. For example, a phase II trial (NCT04701307) for the treatment of SCLC and other NETs is examining the combination of Niraparib, a PARP inhibitor, with Dostarlimab, a biological agent. Other combinations under study are surufatinib–tislelizumab (phase Ib/II—NCT04579757), everolimus–lenvatinib (phase II—NCT03950609), everolimus–sorafenib (phase I—NCT00942682) and everolimus–pasireotide (phase I—NCT00804336) for the treatment of advanced NETs and unresectable types. Telotristat, an anti-diarrheal with anti-neoplastic properties, is being evaluated as an adjunct to Lutathera to determine whether it increases its efficacy (phase II—NCT04543955). Another trial (phase II—NCT00780663) is evaluating the therapeutic role of Quarfloxin, a fluoroquinolone, with anti-neoplastic effects in NETs. The radiopharmaceutical agent I-131 metaiodobenzylguanidine (MIBG) is, likewise, under study for its tumor-shrinking effect in NETs (phase III—NCT00037869 and NCT01099228). The results of these clinical trials are awaited and will hopefully provide a paradigm shift in the management of NEN in the near future.

With regard to newer drug targets, a promising finding has been the presence of tumor-infiltrating lymphocytes (TILs) in pancreatic NETs to be a predictor of survival; hence, modulating the TIL density may add a layer of therapy [204]. Newer targets, like programmed death ligand-1 (PDL1) expression and cytotoxic T lymphocyte antigen-4, have been reported in SCLC and melanoma, whereas PDL-1 has been observed to be an independent prognostic marker in NETs [20,205,206,207]. These findings may help level the ground for future studies, supporting the trials of newer therapies to ensure a better prognosis and quality of life for NET patients.

The current management strategies for various NENs and evolving evidence for their management are summarized in Table 2.

### 12.4. Supportive Management

Effective treatment is not just about treating the disease but is also about treating the patient as a whole. In addition to medical complications faced by an individual diagnosed with NET, other aspects such as physical and mental health disruption, emotional setbacks, financial setbacks and social challenges are also present. This is where supportive care can play an important role. Along with medications to slow down or eradicate the tumor, supportive care is a critical component of NET’s treatment. Furthermore, the management of hormone-related clinical syndromes and paraneoplastic illnesses, or their avoidance, forms part of the targeted supportive care provided to people with NETs. Indeed, supportive therapy is essential in the effective treatment of NET. Examples of supportive therapy include debulking surgery, interventional radiologic techniques to decrease tumor bulk or load, systemic pharmaceutical treatment choices to control or preclude hypersecretion syndromes and treatment-related ill effects.

Supportive therapies add to comprehensive treatment, dealing with the patient as a whole, during the course of the NET treatment procedure. Supportive therapy for cancer patients includes pain management, specialized nursing and psychosocial support. It focuses on assisting patients and their families with non-medical issues in enhancing their quality of life when receiving medical care. Regardless of age, tumor type or stage, everyone can receive this type of care, it is most effective when started as soon as a NET diagnosis is confirmed. People with tumors who also receive supportive and palliative care generally experience less severe symptoms, a higher quality of life and are happier with their treatment [209].

## 13. Conclusions and Future Direction

NENs, besides being a heterogenous group of clinically relevant tumors, also serve as a topic of a spectrum of further study including the histological features and, principally, their management. The involvement of widely distributed neuroendocrine cells aids in the diagnosis due to the region-specific clinical features and specific hormone over secretion-related clinical features, but most tumors remain undiagnosed until a later stage. Their potential to metastasize, their varying histopathology, their presentation as a part of genetic syndromes and their symptomatic treatments add to the confusion of early diagnosis. Recent updates in our knowledge of the histological aspects of these tumors, in terms of genetic factors and pathology, including the use of newer terminology, may help in providing better diagnosis and the characterization of their tumor stage and grade and, thus, better utilization of management options. The development of newer and evolving treatment options based on an improved understanding at their molecular level, including mTOR, VEGF and PRRT, adds to evidence supporting the further possibility of in-depth study and identification of more treatment options beyond the complete resection for these tumors.

## Figures and Tables

**Figure 1 jcm-12-05138-f001:**
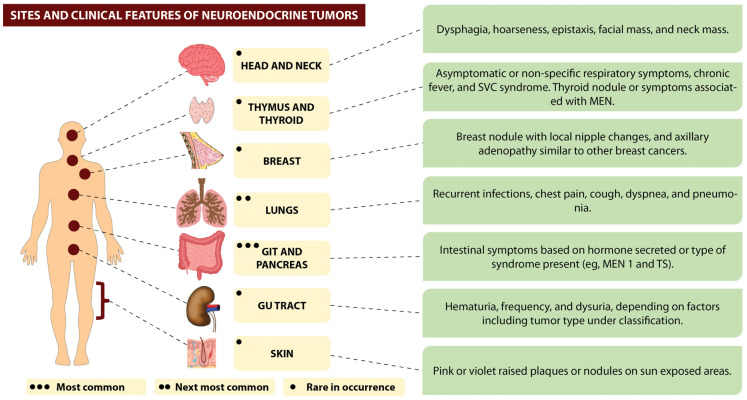
Common locations and clinical features of neuroendocrine tumors. SVC—superior vena cava syndrome; MEN 1—multiple endocrine neoplasia, type 1. *Figure credit: segments of the figure were generated by making use of pictures available from Servier Medical Art, accessed from Servier, and licensed under a Creative Commons Attribution 3.0 unported* Published with permission from PearResearch.

**Figure 3 jcm-12-05138-f003:**
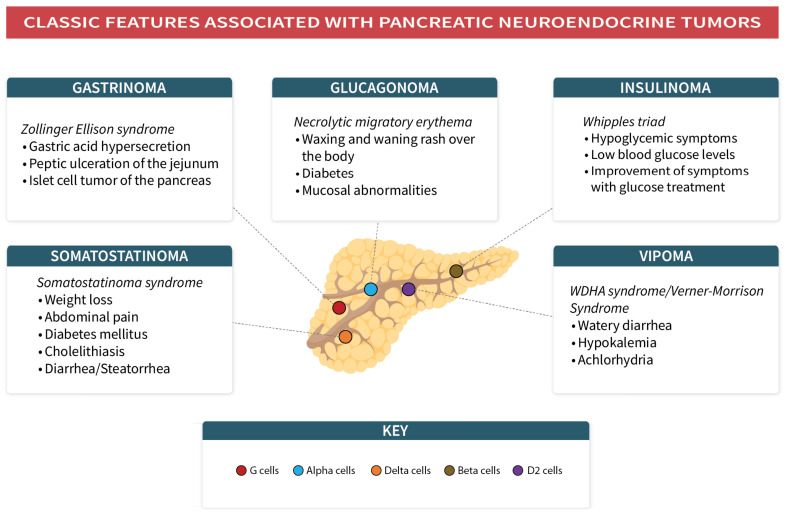
Classic symptoms associated with pancreatic NET. WDHA: watery diarrhea, hypokalemia and achlorhydria; D2 cells: dopamine-D2 cells. *Figure credit: segments of the figure were generated by making use of pictures available from Servier Medical Art, accessed from Servier, and licensed under a Creative Commons Attribution 3.0 unported license* Published with permission from PearResearch.

**Figure 5 jcm-12-05138-f005:**
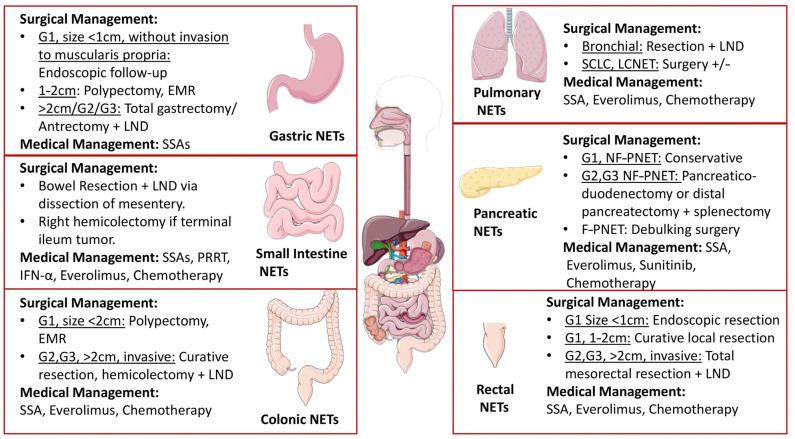
Treatment for various NETs. LND: lymph node dissection, EMR: endoscopic mucosal resection, SSA: somatostatin analog, PRRT: peptide receptor radionuclide therapy, IFN: interferon, NF-NET: non-functioning neuroendocrine tumor, LCNEC: large cell neuroendocrine carcinoma, SCLC: small cell lung carcinoma, F-NET: functioning neuroendocrine tumor. TME: total mesorectal excision, TEM: transanal endoscopic microsurgery, LAR: low anterior resection, ESD: endoscopic submucosal dissection. *Figure credit: segments of the figure were generated by making use of pictures available from Servier Medical Art, accessed from Servier, and licensed under a Creative Commons Attribution 3.0 unported license.* Published with permission from PearResearch.

**Figure 6 jcm-12-05138-f006:**
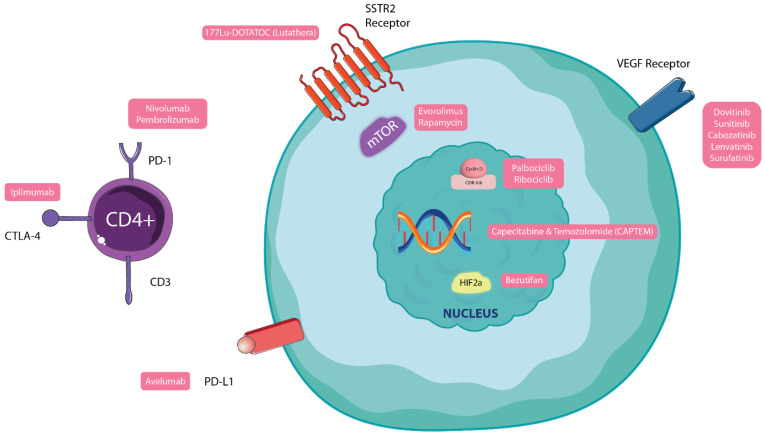
Potential drug targets for NEN and the important drugs. CDK: cyclin-dependent kinase; VEGF: vascular endothelial growth factor; mTOR: mammalian target of rapamycin; HIF: hypoxia-inducible factor. *Figure credit: parts of the figure were generated by making use of pictures available from Servier Medical Art, accessed from Servier, and licensed under a Creative Commons Attribution 3.0 unported license* Published with permission from PearResearch.

**Table 1 jcm-12-05138-t001:** The 2022 World Health Organization (WHO) epithelial neuroendocrine neoplasms classification (8).

Neuroendocrine Neoplasm	Classification Category	Mitotic Count	Ki-67 Index	Other Features
**1. Gastrointestinal tract and pancreato-biliary tract neoplasms**				
a. Well-differentiated neuroendocrine neoplasm (NET)	Grade 1, NET	Less than 2 mitoses/2 mm^2^	Less than 3%	-
	Grade 2, NET	2 to 20 mitoses/2 mm^2^	3 to 20%	-
	Grade 3, NET	More than 20 mitoses/2 mm^2^	More than 20%	-
b. Poorly differentiated neuroendocrine carcinoma (NEC)	Small cell NECs	More than 20 mitoses/2 mm^2^	More than 20% (often more than 70%)	Small cell cytomorphology
	Large cell NECs	More than 20 mitoses/2 mm^2^	More than 20% (often more than 70%)	Large cell cytomorphology
**2. Upper airway, digestive tract and salivary gland neoplasms (head and neck)**				
a. Well-differentiated neuroendocrine neoplasm (NET)	Grade 1, NET	Less than 2 mitoses/2 mm^2^	Less than 20%	No necrosis
	Grade 2, NET	2–10 mitoses/2 mm^2^	Less than 20%	Necrosis
	Grade 3, NET	More than 10 mitoses/2 mm^2^	More than 20%	-
b. Poorly differentiated neuroendocrine carcinomas (NECs)	Small cell NECs	More than 10 mitoses/2 mm^2^	More than 20% (often more than 70%)	Small cell cytomorphology
	Large cell NECs	More than 10 mitoses/2 mm^2^	More than 20% (often more than 55%)	Large cell cytomorphology
**3. Thymus and lung neoplasms**				
a. Well-differentiated neuroendocrine tumor (NET)	Grade 1, typical carcinoid/NET	Less than 2 mitoses/2 mm^2^	-	No necrosis seen
	Grade 2, atypical carcinoid/NET	2 to 10 mitoses/2 mm^2^	-	Necrosis (usually punctate)
	Carcinoids/NETs with mitotic counts and/or Ki-67 proliferation index on the higher side	More than 10 mitoses per 2 mm^2^	More than 30%	Atypical carcinoid morphology
b. Poorly-differentiated neuroendocrine carcinomas (NECs)	Small cell (lung) carcinomas	More than 10 mitoses/2 mm^2^	-	Necrosis and small cell cytomorphology often seen
	Large cell NECs	More than 10 mitoses/2 mm^2^	-	Virtually almost always large cell cyto-morphology and necrosis seen
**4. Thyroid neoplasms**				
a. Medullary thyroid carcinomas (MTCs)	MTC, low-grade	Less than 5 mitoses/2 mm^2^	-	No necrosis seen
	MTC, high-grade	More than or equal to 5 mitoses/2 mm^2^	More than or equal to 5% OR	Necrosis

**Table 2 jcm-12-05138-t002:** A summary of management strategies for various NENs.

S.N.	Type of NEN	Clinical Diagnosis	Lab Diagnosis	Imaging	Treatment Options	Upcoming Evidence/Clinical Trials/Evolving Drug Targets
1.	Pulmonary NENs	**Central tumors**: cough, hemoptysis and recurrent pneumonia. **Peripheral tumors**: non-specific symptoms and complications including, dysphagia, SVC syndrome and hoarseness of voice [4,30,34,35,36].	High levels of ACTH or ADH AND 5 HT [153].	**CT scan**: lesions around the central bronchi in typical carcinoids and SCLC, and peripheral lesions in atypical carcinoids and LCNEC. Calcifications present. **Octreotide scan (111In-pentetreotide)** detects both TC and AC [33].	**Medical**: with somatostatin analogs(SSA), everolimus, PRRT, and chemotherapy [147,192,193,194,195]. **Surgical**: with bronchial resection plus LND [33].	Identification of newer targets including the stimulator of interferon gene (STING) pathway [196], mutations in the epithelial growth factor receptor (EGFR), Phosphoinositol-3 kinase (PI3K), mTOR, VEGF and human epidermal growth factor receptor 2 (HER-2) [197].
2.	GIT NENs	**Gastric**: heartburn, peptic ulcers and diarrhea [28,29]. **Duodenal**: abdominal pain, diarrhea, gastrointestinal bleeding and jaundice [43]. **Appendiceal**: abdominal pain [46]. **Rectal**: abdominal pain, hematochezia, change in bowel movements and pruritis [49,50].	Gastrin/somatostatin/histamine/peptide YY positive [153].	Endoscopy with biopsy and endoscopic ultrasound are required for diagnosis and staging. CT scan and MRI are performed to identify distant metastasis [208].	**Medical**: with SSA, PRRTs, everolimus, IFN alpha chemotherapy [131,157,161,167,168,169]. **Surgical**: with bowel resection + LND, hemicolectomy if ileal involvement [165,166].	Currently, registered clinical trials are aimed at understanding the efficacy of already available treatments in different combinations and comparisons between them, for instance, PRRT vs. temozolomide/capecitabine or everolimus, or sunitinib. Sequential treatments are also being compared [202].
3.	Genitourinary NENs	**Females**: abnormal uterine bleeding is seen in cervical and endometrial tumors [57,58,59,60,61]. **Males**: urinary frequency, urgency and hematuria in prostate tumors. Testicular mass with or without pain. **Bladder tumors**: cause hematuria or urinary obstruction [55,56,60]. **Renal NENs**: Flank pain and hematuria [56,57].	CgA or Synaptophysin or NSE positive [153].	CT scan, MRI and octreotide scan are required for diagnosis and surveillance of lesions. FDG-PET is useful for staging [61].	**Medical**: unresectable tumors are treated with radiotherapy and platinum-based chemotherapy. **Surgical**: for resectable tumors, oncological resection with neoadjuvant chemotherapy [56,57,60,61].	A new combination of chemotherapy agents, BXCL701-Pembrolizumab, is being studied for efficacy in small cell NETs of prostate and adenocarcinoma phenotype (phase Ib/II—NCT03910660) [204].
4.	Pancreatic NENs	**Insulinomas**: tremors, palpitations, diaphoresis, syncope, confusion, anxiety, visual changes and coma [62,76]. **Glucagonomas**: migratory erythema diarrhea, weight loss, diabetes mellitus, DVT, anemia, and stomatitis [83,85]. **Gastrinoma**: Heartburn, abdominal pain, diarrhea and weight loss. Strictures, bleeding and perforation [93]. **Somatostatinomas**: Weight loss, abdominal pain, diarrhea or statorrhea, diabetes mellitus and cholelithiasis [100]. **VIPomas**: Watery diarrhea, hypocalcemia, flushing and glucose intolerance [103].	Insulin levels in the blood must be greater than or equal to 36 pmol/L [78]. Glucagon level of more than 500 to 1000 pg/mL (normal 50 to 150 pg/mL) [62]. Fasting serum gastrin level of 1000 pg/mL [93]. A fasting plasma hormone concentration of more than 3-fold the normal [62]. VIP elevations (>200 pg/dL) are present [104].	Demarcated hyper or hypovascular homogenous tumor on CT scan and endoscopic ultrasound [70].	**Medical**: with SSA, everolimus, sunitinib and chemotherapy [157,181,184]. **Surgical**: G1 via pancreaticoduodenectomy, pancreatectomy + splenectomy pr debulking surgery [176,177,178].	Research is being performed to determine the efficacy of 177Lu-DOTATOC (Lutathera), a new therapeutic agent for PRRT in gastroenteropancreatic NETs. Various phase 2 and phase 3 clinical trials for different combination therapies for pancreatic NETs are also in place [204]. The presence of tumor-infiltrating lymphocytes (TILs) in pancreatic NETs is thought to be a predictor of survival; hence, modulating the TIL density may be another treatment option [204].
5.	Rare NENs	**Head and neck NENs**: dysphagia, hoarseness of voice and epistaxis [114]. **Thymus NENs**: cough, dyspnea, chest pain, weight loss and chronic fever [116]. **Thyroid NENs**: thyroid nodule and cervical lymphadenopathy [117]. **Breast NENs**: solitary breast nodule, axillary lymphadenopathy, nipple discharge and retraction [118]. **Skin NENs**: Warts or blister [119].	CgA or Synaptophysin or NSE positive [153].	CT scan and MRI help identify adrenal lesions and FDG-PET scan distinguishes malignant from benign lesions.	Based on the site of involvement the tumor is managed and some tumors can also present as part of genetic syndromes and are treated with them.	Newer targets: programmed death ligand-1 (PDL1) expression and cytotoxic T lymphocyte antigen-4 are being reported in SCLC and melanoma [20,205,206,207].

SVC: superior vena cava syndrome, ACTH: adrenocorticotrophic hormone, ADH: antidiuretic hormone, 5-HT: 5 Hydroxytryptamine, CT: computed tomography, SCLC: small cell lung cancer, LCNEC: large cell neuroendocrine carcinoma, TC: typical carcinoid, AC: atypical carcinoid, PRRT: peptide receptor radionucleotide therapy, LND: lymph node dissection, EGFR: epidermal growth factor receptor, HER-2: human epidermal growth factor receptor 2, MRI: magnetic resonance imaging, IFN: interferon, CgA: chromogranin A, NSE: neuron-specific enolase, FDG-PET: fludeoxyglucos-positron emission tomography, DVT: deep vein thrombosis, VIP: vasoactive intestinal peptide, G1: Grade 1.

## Data Availability

Not applicable.

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
