# Peer review of "A Comprehensive Review on Neuroendocrine Neoplasms: Presentation, Pathophysiology and Management"

_jcm, 2023, doi:10.3390/jcm12155138_

Round 1

Reviewer 1 Report

Thank you for a comprehensive review. This is a large topic, maybe it is too much for one review paper. 

1. Please describe the NEC morphology more extensively. 

2. MiNEN - minimum 30 % of each component, please clarify.

3. Appendiceal NET. Please include the new paper by Nesti et al and discuss treatment according to that study. Also include a few remarks on L tumors. 

4. P-NET. I prefer functioning and non-functioning tumor, please change accordingly. 

5. Why is MEN-1 not described? Please do so. 

6. The treatment paragraph is hard to read as different tumors are described in one paragraph. Please make smaller paragraphs. 

7. Surgical treatment in selected patients with disseminated Si-NET. Is that a possibility, please describe if yes. 

8. Is the NETTER study described? And could you pelase elaborate why it is an important recent study. 

Author Response

The Editor-in-Chief,

JCM-2531268 (MDPI)

Dear Editor,

Thank you for moving our manuscript forward into the peer review process and providing us the Reviewers’ reports. We appreciate the time and effort that the Reviewers kindly provided in making suggestions to improve our manuscript. We have carefully considered all their comments and have made required corrections into our manuscript using the ‘track changes’ function, as informed in our previous communication.

We have addressed the comments made in relation to the manuscript as follows:

Reviewer 1

Thank you for a comprehensive review. This is a large topic, maybe it is too much for one review paper. 

We express our sincere thanks to Reviewer 1 for their detailed evaluation. We are pleased that the Reviewer valued our work and appreciated our intent to comprehensively compile data on NEC into a single article, as, to the best of our knowledge, no single review has so extensively compiled this literature, synthesizing and simplifying it with supportive illustrations. We have revised our manuscript per your comments as follows:

  1. Please describe the NEC morphology more extensively. 

Authors’ reply: Thank you for the comment. We have extended the Introduction to highlight the importance of morphology – by giving a brief description. The individual morphologies have already been described under the sections describing the individual tumors.

  1. MiNEN - minimum 30 % of each component, please clarify.

Authors’ reply: Thank you for pointing out the inconsistency. We have rephrased and replaced the ambiguous text within our revised manuscript.

  1. Appendiceal NET. Please include the new paper by Nesti et al and discuss treatment according to that study. Also include a few remarks on L tumors. 

Authors’ reply: Thank you for helping us with the recent Nesti et al. update. We have included it in the text of our revised article.

  1. P-NET. I prefer functioning and non-functioning tumor, please change accordingly. 

Authors’ reply: Thank you for pointing this out. It has been corrected as per your suggestion.

  1. Why is MEN-1 not described? Please do so. 

Authors’ reply: Thank you for pointing that out. We have now described it under section 7.4.

  1. The treatment paragraph is hard to read as different tumors are described in one paragraph. Please make smaller paragraphs. 

Authors’ reply: Thank you for pointing that out. We have revised and restructured this paragraph for better flow and readability.

  1. Surgical treatment in selected patients with disseminated Si-NET. Is that a possibility, please describe if yes.

Authors’ reply: Yes, but going into any significant detail of this is beyond the scope of the present article (which is already towards the upper limit of the journal’s suggested length). We have touched on this matter under section 12.3. Please find references attached:

https://www.ncbi.nlm.nih.gov/pmc/articles/PMC7408509/

https://www.ncbi.nlm.nih.gov/pmc/articles/PMC9131835/

  1. Is the NETTER study described? And could you pelase elaborate why it is an important recent study. 

Authors’ reply: We have described the trial under section 12.3. We have now included the published study too and expanded our explanation. Thank you for pointing this out.

Reviewer 2 Report

I’ve carefully read the manuscript entitled “A Comprehensive Review on Neuroendocrine Neoplasms: Presentation, Pathophysiology, and Management” which is indeed an extensive review on neuroendocrine lesions. 

The review is well structured and types of lesions according to anatomical distribution are approached in detail. Figures are also representative. From a clinical point of view, the paper is very informative for the practicing physician, covering all types of neuroendocrine tumors. Details about diagnosis of NEN are less represented, and particularly for pancreatic NEN, EUS should more discussed, including its therapeutic role (syst review by Armellini et al). 

The authors should highlight the added value of their paper in the current literature, as there are already several reviews on the topic already published - discuss the diagnostic difficulties and delay, the wide spectrum of clinical features and the need for a high clinical suspicion.

Author Response

Reviewer 2

I’ve carefully read the manuscript entitled “A Comprehensive Review on Neuroendocrine Neoplasms: Presentation, Pathophysiology, and Management” which is indeed an extensive review on neuroendocrine lesions. 

The review is well structured and types of lesions according to anatomical distribution are approached in detail. Figures are also representative. From a clinical point of view, the paper is very informative for the practicing physician, covering all types of neuroendocrine tumors. Details about diagnosis of NEN are less represented, and particularly for pancreatic NEN, EUS should more discussed, including its therapeutic role (syst review by Armellini et al). 

The authors should highlight the added value of their paper in the current literature, as there are already several reviews on the topic already published - discuss the diagnostic difficulties and delay, the wide spectrum of clinical features and the need for a high clinical suspicion.

Authors’ reply: We are delighted that Reviewer 2 appreciated our work. Thank you for helping us with detailed feedback. We have included significant information from your suggested study into our revised manuscript. We have also slightly expanded our Introduction to underline the rationale for our review article. Thank you again for helping us improve our article.

In closure, all authors express their sincere thanks to the Respected Editor and both Reviewers for their help and time in improving our revised article.

With best regards - the corresponding authors.

Round 2

Reviewer 1 Report

Tahnk you very much for a very thorough revision. I am pleased and have no further comments. 

Author Response

The authors wish to thank both Reviewers 1 and 2 for their valuable suggestions, as well as for the time that both kindly took in reading and reviewing our article